# A Rho-actin signaling pathway shapes cell wall boundaries in *Arabidopsis* xylem vessels

Yuki Sugiyama[1,2], Yoshinobu Nagashima[1,2], Mayumi Wakazaki[3], Mayuko Sato[3], Kiminori Toyooka[3], Hiroo Fukuda[1] & Yoshihisa Oda [2,4]

Patterned cell wall deposition is crucial for cell shapes and functions. In Arabidopsis xylem vessels, ROP11 GTPase locally inhibits cell wall deposition through microtubule disassembly, inducing pits in cell walls. Here, we show that an additional ROP signaling pathway promotes cell wall growth at pit boundaries. Two proteins, Boundary of ROP domain1 (BDR1) and Wallin (WAL), localize to pit boundaries and regulate cell wall growth. WAL interacts with F-actin and promotes actin assembly at pit boundaries while BDR1 is a ROP effector. BDR1 interacts with WAL, suggesting that WAL could be recruited to the plasma membrane by a ROP-dependent mechanism. These results demonstrate that BDR1 and WAL mediate a ROP-actin pathway that shapes pit boundaries. The study reveals a distinct machinery in which two closely associated ROP pathways oppositely regulate cell wall deposition patterns for the establishment of tiny but highly specialized cell wall domains.

[1] Department of Biological Sciences, Graduate School of Science, The University of Tokyo, 7-3-1 Hongo, Bunkyo-ku, Tokyo 113-0033, Japan. [2] Center for Frontier Research, National Institute of Genetics, 1111 Yata, Mishima, Shizuoka 411-8540, Japan. [3] RIKEN Center for Sustainable Resource Science, RIKEN, 1-7-22 Suehiro, Tsurumi, Kanagawa, Yokohama 230-0045, Japan. [4] Department of Genetics, SOKENDAI (Graduate University for Advanced Studies), 1111 Yata, Mishima, Shizuoka 411-8540, Japan. These authors contributed equally: Yuki Sugiyama, Yoshinobu Nagashima. Correspondence and requests for materials should be addressed to Y.O. (email: oda@nig.ac.jp)

Rho GTPases regulate the behavior of the cytoskeleton through various cellular events[1,2]. In plants, Rho-like GTPases from plant (ROP) control cell wall deposition pattern by governing the behavior of microtubules[3–6] and actin filaments[6–14], which thereby determines cell shapes and functions[15–17]. However, how specialized domains are established in cell walls with edges and boundaries through the action of ROP signaling remains poorly understood. During the development of xylem vessels, cell wall deposition is locally inhibited to form pits in secondary cell walls, through which water moves between xylem vessels. Rho-like GTPase from plant 11 (ROP11) is locally activated to induce microtubule disassembly through its effector, MIDD1, and Kinesin-13A, resulting in the formation of pits[18–21]. During pit formation, bordered cell walls specifically develop at the boundary of pits. However, little is known about how the distinct boundaries of pits are established along with ROP11-MIDD1-dependent pit formation.

In this study, we show that an additional ROP signaling pathway promotes cell wall growth at pit boundaries. Two proteins, boundary of ROP domain1 (BDR1) and wallin (WAL), localize to pit boundaries and regulate cell wall growth. WAL interacts with F-actin and promotes actin assembly at pit boundaries, while BDR1 is found to be a ROP effector. BDR1 interacts with WAL, suggesting that WAL could be recruited to the plasma membrane by a ROP-dependent mechanism. These results demonstrate that BDR1 and WAL mediate a ROP-actin pathway that shapes pit boundaries.

## Results

**WAL promotes cell wall ingrowth at pit boundaries.** To identify potential factors connecting ROP11 signaling with the pit boundary, uncharacterized genes that were upregulated during metaxylem vessel differentiation in *Arabidopsis*[22] were identified, and the subcellular localization of their products in metaxylem vessel cells was investigated[23]. A protein, designated WAL, was identified that localized to filamentous structures accumulating at the boundary of secondary cell wall pits (Supplementary Figure 1). A *pWAL:GFP-WAL* construct was expressed in xylem vessels, and was found to localize at pit boundaries in roots (Fig. 1a). *pMIDD1:MIDD1ΔN-tagRFP*, a marker for the ROP11-MIDD1 pathway at pits, was introduced into *pWAL:GFP-WAL* plants. GFP-WAL localized at the edges of MIDD1ΔN-tagRFP domains, indicating that WAL localized at pit boundaries (Fig. 1b). *WAL* messenger RNA (mRNA) levels in *wal* plants, in which T-DNA was inserted into an exon at the *WAL* locus (SAIL_729_H08), were ~10% of those in wild-type plants (Fig. 1c). The *wal* plant displayed larger and rounder secondary cell wall pits in metaxylem vessels than did wild-type plants. *pWAL:GFP-WAL* fully complemented the pit phenotype of *wal*, indicating that WAL was required for proper pit structure (Fig. 1d, e). Fine pit structure was examined using electron microscopy (Fig. 1f). Wild-type plants formed typical bordered cell walls, in which cell walls overarched the pit membrane to produce narrow pit apertures. By contrast, *wal* plants failed to form the cell wall arch, resulting in expanded pit apertures. Pit structure was quantitatively evaluated using confocal microscopy (Fig. 1g). Pit aperture in *wal* plants was ~1.8-fold wider than in wild-type plants, but pit membrane width was comparable between *wal* and wild-type plants (Fig. 1h). These data suggested that *WAL* promoted cell wall ingrowth over the pit membrane.

**WAL promotes actin assembly at pit boundaries.** *WAL* was found to encode a protein of unknown function with a predicted short coiled-coil domain (Fig. 2a). TagRFP-WAL was ectopically expressed in non-xylem epidermal cells of *Nicotiana*

*benthamiana* leaves to identify their interacting components (Fig. 2b). WAL co-localized with actin microfilaments labeled with GFP-Lifeact. Truncated WAL lacking the C-terminal half (WALΔC) also localized to actin microfilaments, while WAL lacking the N-terminal half (WALΔN) localized to the plasma membrane but not to actin microfilaments (Fig. 2b), indicating that the N-terminal half of WAL was required for actin localization. During an in vitro co-sedimentation assay, GST-WALΔC was predominantly found in the pellet in the presence of F-actin, but was found in both the pellet and the supernatant in the absence of F-actin. This suggests that WAL binds directly to F-actin (Fig. 2c).

To examine the involvement of actin in bordered cell wall formation, *Lifeact-tagRFP* under the control of the *MIDD1* promoter, which specifically directs gene expression in xylem vessels[23], was introduced into *pWAL:GFP-WAL* plants (Fig. 2d). Actin microfilament networks in xylem cells were observed in both wild-type and *wal* plants (Fig. 2e). Actin microfilament rings that co-localized with GFP-WAL (Fig. 2d) were observed frequently in wild-type plants but rarely in *wal* plants (>90% in wild-type vs. 10% in *wal*) (Fig. 2e, f). Treatment for 2 days with latrunculin B, an F-actin disruption drug, produced pits that were larger and rounder and lacked the cell wall arch (Supplementary Figure 2d), resembling pits seen in *wal* plants. These results suggest that WAL regulates pit structure through formation of an actin ring.

Transient treatment with latrunculin B for 6 h (rather than 2 days) was used to investigate actin ring properties without affecting cell wall structure (Fig. 2g, h). Cytoplasmic actin filaments completely disassembled after treatment with latrunculin B, but actin rings persisted at the pits, suggesting that actin filaments at the pits were more stable than those in the cytoplasm. This suggests that WAL stabilizes actin filaments to allow formation of an actin ring.

GFP-tagged truncated WAL proteins were observed in *wal* plants to investigate WAL localization. GFP-WALΔN localized at pit boundaries, resembling the localization of untruncated GFP-WAL (Supplementary Figure 2a and c). GFP-WALΔC localized to actin-like filaments (Supplementary Figure 2b), suggesting that the C-terminal domain of WAL was required for pit-specific localization of WAL. WAL may be recruited to pits through its C-terminal domain by unknown proteins.

**BDR1 recruits WAL to pit boundaries.** To identify the potential factors that recruit WAL to pits, the localization of uncharacterized xylem-expressed gene products was further investigated in cultured cells. A DUF620 domain-containing protein, termed BDR1, localized at secondary cell wall pit boundaries (Supplementary Figure 3a). *Arabidopsis* has eight additional *DUF620* genes, *BDR2–BDR9* (Supplementary Figure 3b). Expression data[24] indicated that *BDR1*, *BDR2*, *BDR3*, and *BDR4* were expressed in xylem-related tissues (Supplementary Figure 3c). In roots, BDR1 localized to the boundaries of pit membranes labeled with MIDD1ΔN (Fig. 3a) and co-localized with WAL (Fig. 3b). Mutant *bdr1-1* plants, in which no *BDR1* mRNA was detected due to a T-DNA insertion into a *BDR1* exon (Supplementary Figure 3d), exhibited no aberrant pit structure phenotype. However, *bdr1-1 BDR3RNAi* plants, in which *BDR3* mRNA levels were reduced to 45% of those of the wild type upon treatment with estrogen (Supplementary Figure 3d), exhibited larger and rounder pits than wild-type plants (Fig. 3c, d), suggesting that *BDR1* and *BDR3* redundantly regulated pit structure.

To determine whether the BDR1-GFP fusion protein was functional, we introduced *pBDR1:BDR1-GFP* into *bdr1-1 BDR3RNAi* plants. Compared to *bdr1-1 BDR3RNAi* plants,

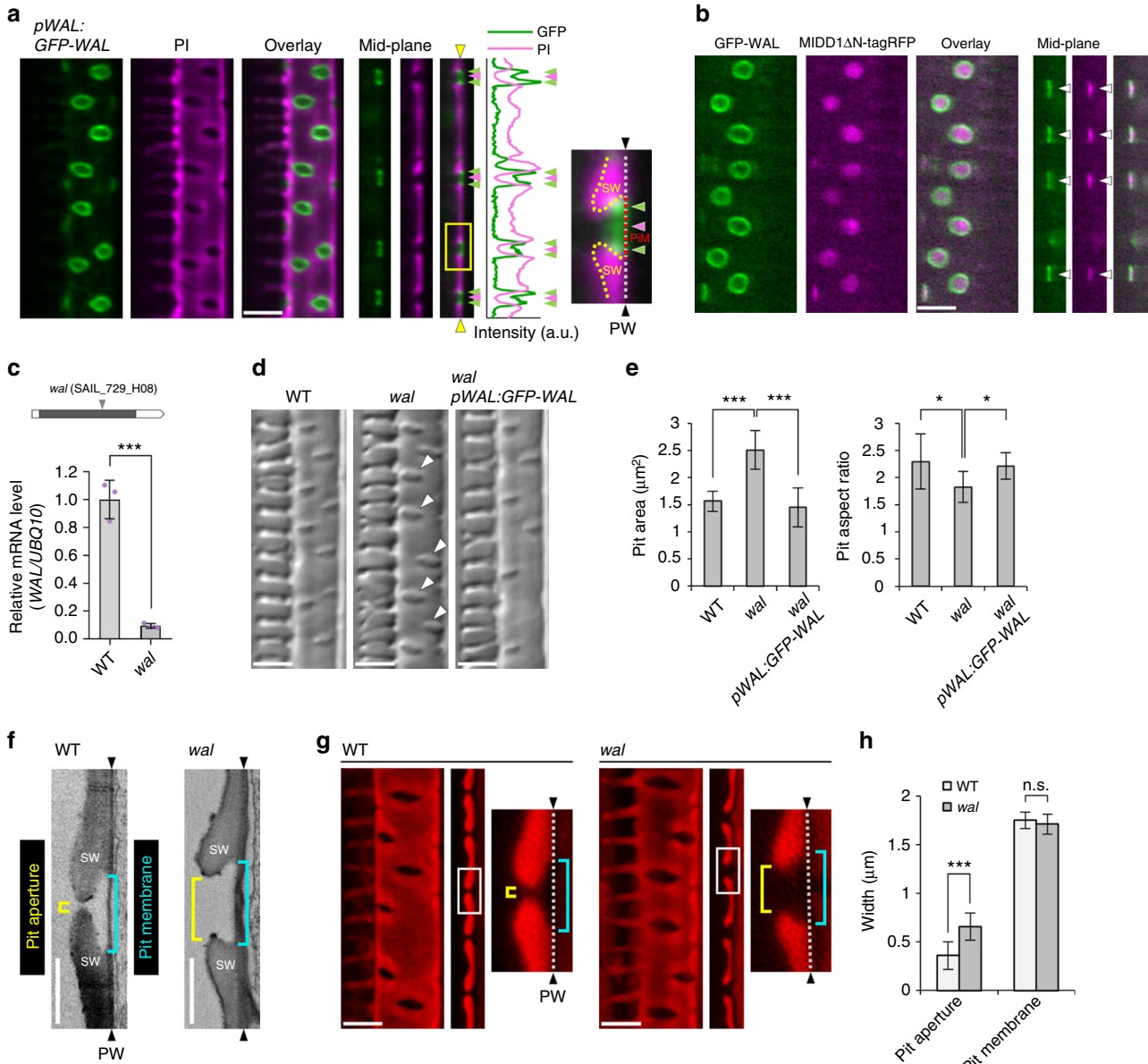

**Fig. 1** WAL is required for cell wall arches of pits. **a** Localization of GFP-WAL (*pWAL:GFP-WAL*) in metaxylem vessel cells in roots. Cell walls are labeled with propidium iodide (PI). Cortex (left) and mid-plane (right) from different cells are shown. The graph shows the intensity profile between the yellow arrowheads. GFP-WAL (green arrowheads) localized to the pit boundaries (red arrowheads). The boxed area is magnified in the right panel. Yellow and red broken lines indicate the edges of secondary cell walls (SW) and pit membranes (PiM), respectively. **b** Localization of GFP-WAL and active ROP11 domains (*pMIDD1:MIDD1ΔN-tagRFP*) in metaxylem vessel cells in roots. Cortex (left) and mid-plane (right) from different cells are shown. **c** *WAL* mRNA abundance in *wal* (SAIL_729_H08) plants. Values are mean ± s.d. (*n* = 3), ****p* < 0.001 (Student's *t*-test). **d** Differential interference contrast (DIC) of xylem vessels in roots of wild-type, *wal*, and *wal pWAL:GFP-WAL* plants. Arrowheads indicate enlarged secondary cell wall pits. **e** Surface area and aspect ratios of secondary cell wall pits. Values are mean ± s.d. (*n* > 300 pits), **p* < 0.05; ****p* < 0.001 (ANOVA with Scheffe test). **f** Electron micrographs of metaxylem vessel cells in roots. Yellow and blue brackets indicate pit aperture and pit membrane, respectively. Black arrowheads indicate the position of the primary cell wall layer. **g** Secondary cell walls of metaxylem vessels in roots of wild-type and *wal* plants. Secondary cell walls were stained with safranin. High-resolution confocal images were acquired with FV-OSR technology. Maximum intensity projection (left) and mid-plane (center) from different cells are shown. Right panels show magnification of the boxed regions. Yellow and blue brackets indicate pit aperture and pit membrane, respectively. White broken lines indicate the position of the primary cell wall layer. **h** Width of pit apertures and pit membranes. Values are mean ± s.d. (*n* > 240 pits), n.s., not significant; ****p* < 0.001 (Student's *t*-test). Bars = 5 μm (**a**, **b**, **d**, and **g**) and 2 μm (**f**). Source data are provided as a Source Data file

*bdr1-1 BDR3RNAi* plants harboring *pBDR1:BDR1-GFP* exhibited a smaller surface area with a higher aspect ratio of pits upon treatment with estrogen (Supplementary Figure 3e and f). The values were comparable to those of wild-type plants (Fig. 3c), indicating that BDR1-GFP complemented the pit phenotype. We

therefore concluded that the BDR1-GFP fusion protein was functional.

Bimolecular fluorescence complementation (BiFC) analysis was performed to test whether BDR interacts with WAL. EYFP signals were detected when nEYFP (N-terminal half of EYFP)-WAL was

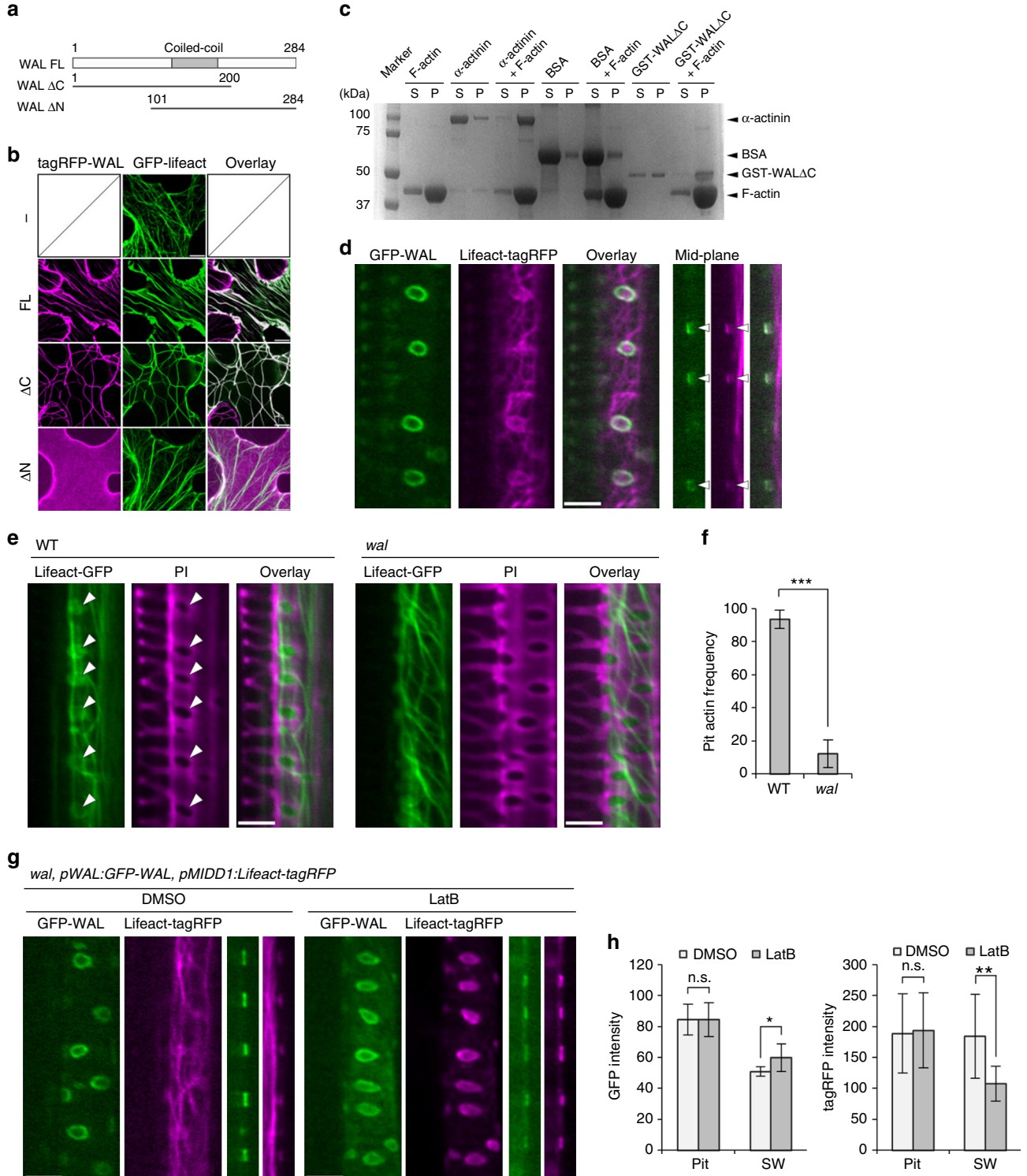

**Fig. 2** WAL is required for actin assembly at pits. **a** Diagram of full-length WAL and truncated WAL fragments. The coiled-coil domain is indicated in gray. Numbers indicate the positions of amino acid residues. **b** tagRFP-fused full-length (FL) and truncated (ΔC and ΔN) WAL and actin microfilaments (*35S: Lifeact-GFP*) in tobacco leaf epidermal cells. **c** F-actin co-sedimentation assay. GST-WALΔC was detected in the pellet (P) but not in the supernatant (S) after co-sedimentation with F-actin. BSA and α-actinin were used as negative and positive controls, respectively. **d** GFP-WAL and actin microfilaments (*pMIDD1:Lifeact-tagRFP*) in metaxylem vessel cells in roots. Cortex (left) and mid-plane (right) from different cells are shown. **e** Localization of actin microfilaments (*pMIDD1:Lifeact-GFP*) in metaxylem vessel cells in roots of wild-type and *wal* plants. Arrowheads indicate actin microfilaments in secondary cell wall pits. **f** Frequency of actin microfilaments in secondary cell wall pits. Values are mean ± s.d. (*n* > 200 pits), ***p < 0.001 (Student's *t*-test). **g** GFP-WAL and actin microfilaments (Lifeact-tagRFP) in metaxylem vessel cells in roots treated with (LatB) or without (DMSO) 30 μM latrunculin B for 6 h. **h** Intensity of WAL and actin microfilaments inside (pit) and outside (SW) secondary cell wall pits. Values are mean ± s.d. (*n* > 150), n.s., not significant; *p < 0.05; **p < 0.01 (*t*-test). Bars = 10 μm (**b**) and 5 μm (**d**, **e**, and **g**). Source data are provided as a Source Data file

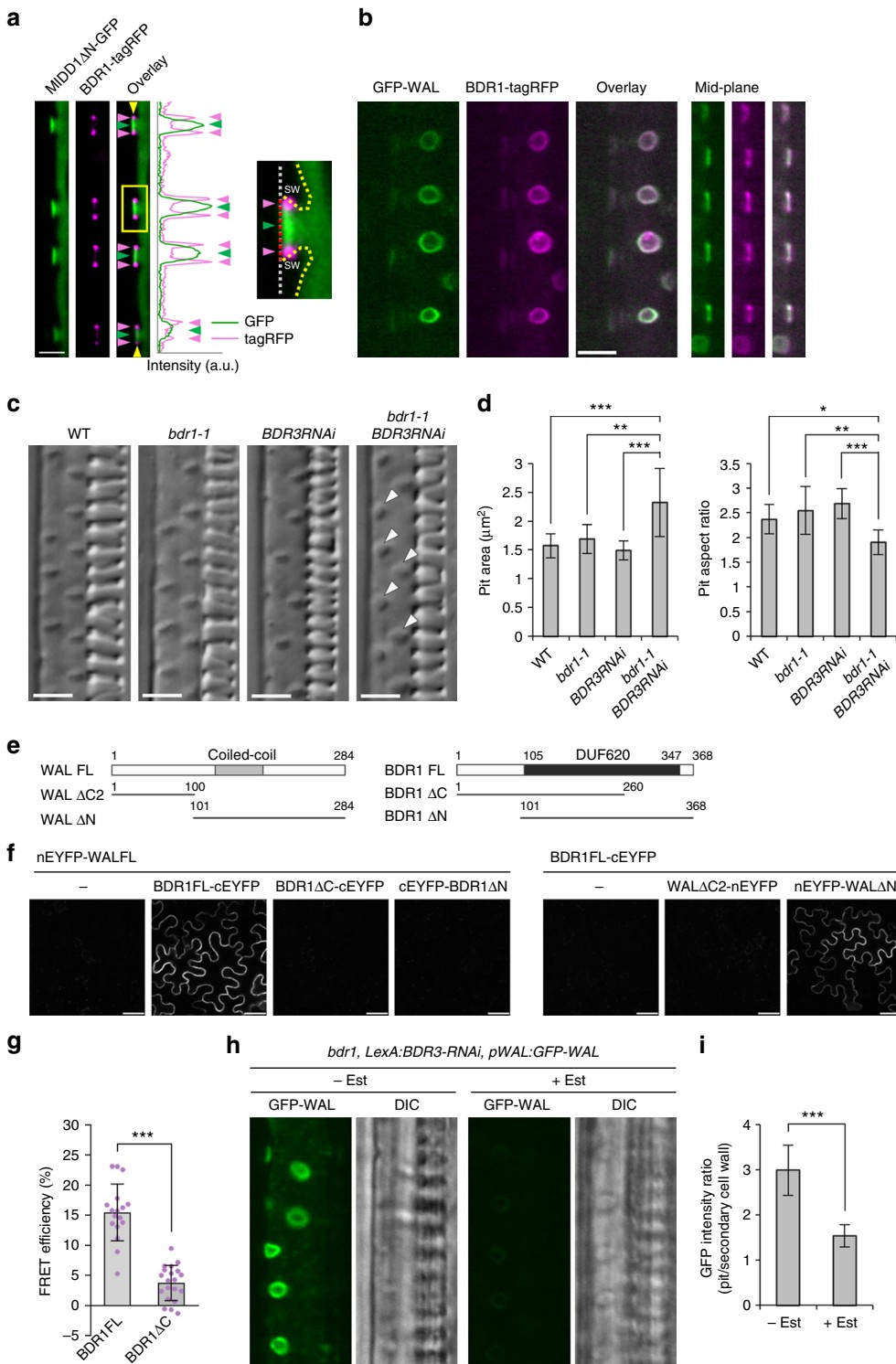

**Fig. 3** BDR recruits WAL to the boundaries of pit membranes. **a** BDR1 (*pBDR1:BDR1-tagRFP*) and MIDD1ΔN (*pMIDD1:MIDD1ΔN-GFP*) in metaxylem vessel cells in roots. The mid-plane is shown. BDR1-tagRFP (red arrowheads) localized to the boundary of the MIDD1ΔN-GFP domain (green arrowheads). The graph shows the intensity profile between the yellow arrowheads. The boxed area is magnified in the right panel. Yellow broken lines indicate edges of secondary cell walls (SW). The red broken line indicates the pit membrane. **b** WAL (*pWAL:GFP-WAL*) and BDR1 (*pBDR1:BDR1-tagRFP*) in metaxylem vessel cells in roots. Cortex (left) and mid-plane (right) from different cells are shown. **c** DIC of WT, *bdr1-1*, *BDR3RNAi*, and *bdr1-1 BDR3RNAi* plants. **d** Surface area and aspect ratio of secondary cell wall pits. Values are mean ± s.d. (*n* > 250 pits), *$p < 0.05$; **$p < 0.01$; ***$p < 0.001$ (ANOVA with Scheffe test). **e** Diagram of full-length and truncated fragments of WAL and BDR1. The black region indicates a DUF620 domain. Numbers indicate the positions of amino acid residues. **f** BiFC assay between WAL and BDR1 in leaf epidermal cells. BiFC signal was observed when nEYFP-WAL FL or nEYFP-WALΔN was co-expressed with BDR1 FL-cEYFP. **g** FRET efficiency between EYFP-WAL and BDR1-ECFP and between EYFP-WAL and truncated BDR1ΔC-ECFP. Values are mean ± s.d. (*n* > 15), ***$p < 0.001$ (Student's *t*-test). **h** GFP-WAL in metaxylem vessel cells in roots of *bdr1-1 BDR3RNAi* plants treated with (+Est) or without (−Est) 2 μM estradiol. **i** Intensity ratio of GFP-WAL (inside of pit/outside of pit). Values are mean ± s.d. (*n* > 30), ***$p < 0.001$ (Student's *t*-test). Bars = 50 μm (**f**) and 5 μm (**a–c** and **h**). Source data are provided as a Source Data file

expressed alongside BDR1-cEYFP (C-terminal half of EYFP), but no signal was observed when either nEYFP-WAL or BDR1-cEYFP was expressed alone (Fig. 3e, f), indicating that WAL interacted with BDR1 in vivo. By contrast, BiFC with a truncated WAL construct (WALΔC2-nEYFP) or truncated BDR constructs (BDR1ΔC-cEYFP or cEYFP-BDR1ΔN) resulted in no EYFP signal, suggesting that the C-terminal half of WAL and both the N- and C-terminal domains of BDR1 were required for interaction between WAL and BDR1.

Fluorescence resonance energy transfer (FRET) analysis in tobacco leaf epidermis confirmed in vivo interaction between WAL and BDR1. FRET efficiency between WAL and BDR1 was 15%, whereas efficiency between WAL and BDR1ΔC was <5% (Fig. 3g). In pull-down assays with purified proteins, GST-WAL and GST-WALΔN were detected in the assay output, whereas GST alone was not detected (Supplementary Figure 3g). These results demonstrate that BDR1 directly interacts with WAL.

Next, pWAL:GFP-WAL was introduced into bdr1-1 BDR3RNAi plants, and WAL-GFP localization was examined after estrogen treatment. Estrogen treatment reduced the intensity ratio (inside pits/outside pits) of GFP-WAL to 50% of that in non-estrogen-treated plants (Fig. 3h, i). Conversely, BDR1-GFP localization was not affected in wal (Supplementary Figure 3h and i). These observations suggest that BDR1 and BDR3 recruit WAL to the pits.

**ROP can recruit WAL to the plasma membrane via BDR1.** ROP11 is activated at the pit membrane[18,19]. BiFC and FRET experiments in tobacco leaves were used to test whether ROP11 interacted with BDR1. EYFP signals were detected between BDR1-nEYFP and cEYFP-ROP11 or cEYFP-ROP11G17V (a constitutive active form), but not between BDR1-nEYFP and cEYFP-ROP11T22N (a constitutive inactive form) (Fig. 4a). FRET efficiencies between BDR1-ECFP and EYFP-ROP11 or between BDR1-ECFP and EYFP-ROP11G17V were >10%, while the efficiency between BDR1-ECFP and EYFP-ROP11T22N was 0% (Fig. 4b). These data strongly suggest that BDR1 is a novel ROP11 effector that binds specifically to the active form of ROP11.

The possibility that active ROP11 might recruit BDR1, and that BDR1 might then recruit WAL in turn, was then tested (Fig. 4c). BDR1-GFP localized to the cytoplasm in cultured non-xylem cells when BDR1-GFP was expressed alone or together with tagRFP-ROP11T22N, whereas BDR1-GFP localized to the plasma membrane when co-expressed with tagRFP-ROP11 or tagRFP-ROP11G17V (Fig. 4c). This suggests that active ROP11 can recruit BDR1 to the plasma membrane.

Next, the ability of active ROP11 to recruit WAL via BDR1 was tested. ROP11 can be activated locally at the plasma membrane by introducing ROP11 together with tagRFP-ROPGEF4PRONE and ROPGAP3 into the tobacco leaf epidermis. ROPGEF4PRONE marks the location of active ROP11[18,25]. BDR1-ECFP or EYFP-fused truncated WAL (WALΔN), which contains the BDR-interacting region but lacks the actin-binding region of WAL, was introduced into tobacco epidermis alongside these components. BDR1-ECFP accumulated around the tagRFP-ROPGEF4PRONE domains, whereas EYFP-WALΔN localized independently of tagRFP-ROPGEF4PRONE. By contrast, when both BDR1-ECFP and EYFP-WALΔN were introduced, EYFP-WALΔN localized around the tagRFP-ROPGEF4PRONE domains together with BDR1-ECFP (Fig. 4d). This demonstrated that active ROP11 recruited BDR1, which in turn recruited WAL to the plasma membrane domains. To determine whether ROP11 could affect the localization of BDR in planta, LexA:GFP-ROP11G17V was

introduced into pBDR1:BDR1-tagRFP plants. Induction for 2 days with estrogen resulted in the loss of pit localization of BDR1-tagRFP and uniform localization of BDR1-tagRFP in xylem vessel cells in association with GFP-ROP11G17V (Supplementary Figure 4), demonstrating that activated ROP11 could direct the localization of BDR1.

## Discussion

Overall, these results indicate that the ROP-BDR-WAL-actin pathway is the central component of the machinery promoting cell wall deposition at pit boundaries (Fig. 4e). ROP11 is activated at the pit membrane to recruit MIDD1[18,19,23]. Considering the ability of BDR1 to interact with the active form of ROP11, ROP11 could be the major candidate for the recruitment of BDR1 to pits. Nevertheless, it is still possible that ROP family members other than ROP11 regulate BDR1 recruitment redundantly or independently of ROP11. In Arabidopsis pavement cells, ROP6 locally promotes parallel microtubule arrays to form indentations, whereas ROP2 and ROP4 promote actin assembly to form robes[5,6]. In root hairs, ROP10 stabilizes the shank of the root hair through microtubules[4], whereas ROP2 regulates tip growth at the apex through actin[10,11]. Similarly, different ROPs may regulate BDR-WAL-actin pathways and the MIDD1-microtubule pathway independently. Further investigation of the behavior of ROPs is needed to determine which ROPs regulate BDR1.

The mechanism by which BDR1 specifically marks the periphery of the pit membrane remains also to be determined. One possibility is that ROP11 and/or other ROPs recruit BDR1 to the pit membrane, but BDR1 is subsequently eliminated from the surface or simply targeted to the boundary of pits by additional machineries. Another possibility is that ROPs are specifically activated at the boundary of nascent pits and specifically recruit BDR1 but not MIDD1.

Nonetheless, this study has identified an additional ROP signaling pathway that specifically shapes pit boundaries. The ROP-BDR-WAL-actin pathway promotes cell wall growth at pit boundaries, while the ROP-MIDD1-microtubule pathway inhibits cell wall deposition at the pit area[18,19,23] (Fig. 4e). The close association of these two signaling pathways, which have opposite effects on cell wall growth, would allow precise control over cell wall deposition in the small pit area. Thus, this study reveals the existence of a distinct machinery for the establishment of tiny but highly specialized cell wall domains.

Pits with bordered cell walls form the major structures in xylem vessels known as bordered pits[26–28]. Bordered pits are essential for robust water transport in plants and limit the spread of air embolisms and vascular pathogens[28,29]. WAL and BDR are conserved in flowering plants (Supplementary Figure 5a and b), and actin rings are also found in the pits in several tree species[30,31], suggesting that the ROP-BDR-WAL pathway may be widely conserved and contribute to efficient and robust water transport in a diverse range of plants.

## Methods

**Plant materials.** Arabidopsis thaliana suspension cells (Col-0) were cultured in 27 ml of modified Murashige and Skoog (MS) medium containing 4.33 g l⁻¹ MS inorganic salts (Wako), 3% sucrose, 1 mg l⁻¹ 2,4-D, and B5 vitamins including 4 mg l⁻¹ nicotinic acid, 4 mg l⁻¹ pyridoxine-HCl, 40 mg l⁻¹ thiamine-HCl, and 400 mg l⁻¹ myoinositol, pH 5.8. Cells were agitated on a rotary shaker at 124 r.p.m. at 22 °C in the dark. At weekly intervals, 12-ml aliquots of the culture were transferred to 15 ml of fresh medium in a 100-ml culture flask[23].

For transformation, 7-day-old cells were co-cultured with Rhizobium radiobacter strain GV3101 MP90 for 48 h in MS medium supplemented with 50 mg l⁻¹ acetosyringone. Claforan (0.5 mg l⁻¹; Aventis) was added to the culture, and the suspension cells were cultured for a further 5 days[23].

To induce xylem differentiation, 1 ml of a 7-day-old transgenic suspension cell culture harboring LexA:VND6 (eVND6 line) was transferred into a 15-ml tube and diluted with 9 ml of MS medium without 2,4-D. Cells were allowed to settle for

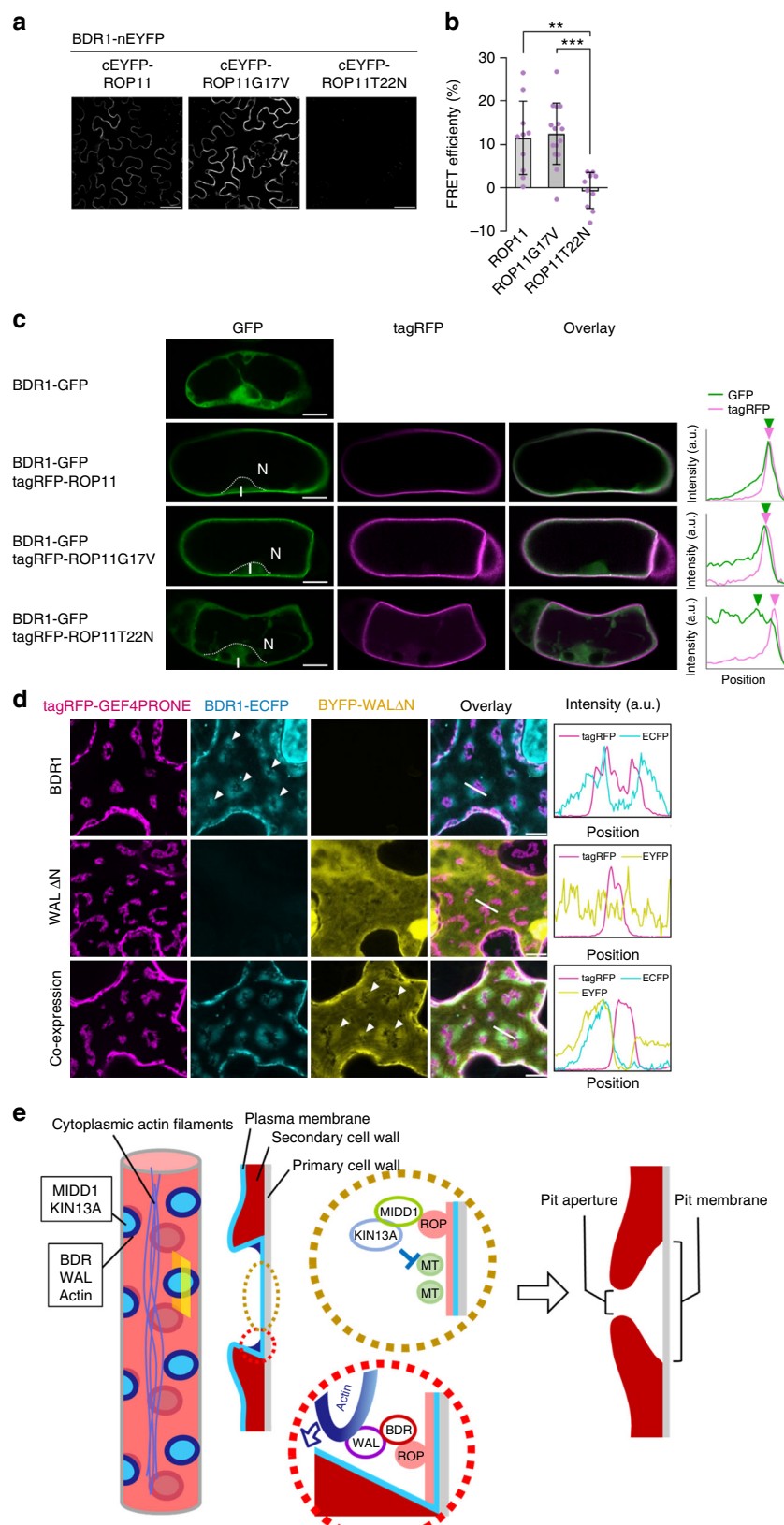

5 min, after which the upper 5 ml of the medium was removed to adjust cell density. The suspension culture was supplied with 2 μM estradiol (10 mM stock in DMSO) and 2 μM brassinolide (10 mM stock in DMSO), and cultured for 24 h[23].

All *A. thaliana* plants used in this study were in the Col-0 ecotype background. T-DNA insertion lines were obtained from Arabidopsis Biological Resource Center (ABRC). Seedlings were grown on 1/2 MS agar medium containing 0.2% sucrose or in soil at 22 °C under constant light. For the expression of estrogen-inducible genes, 5-day-old seedlings were transferred onto 1/2 MS agar medium containing 2 μM estradiol and incubated for 2 days.

**Fig. 4** ROP recruits the BDR-WAL complex to the plasma membrane. **a**, **b** BiFC (**a**) and FRET (**b**) assays between BDR and ROP11, ROP11G17V, or ROP11T22N in tobacco leaf epidermis. Values are mean ± s.d. ($n > 9$), $**p < 0.01$; $***p < 0.001$ (ANOVA with Scheffe test). **c** BDR1-GFP and tagRFP-ROP11 in *Arabidopsis*-cultured cells. N indicates the nucleus. Graphs show intensity profiles along the white lines. BDR1-GFP is localized to the cytoplasm in the absence of ROP11 or in the presence of ROP11T22N, but is localized to the plasma membrane in the presence of ROP11 or ROP11G17V. Green and red arrowheads indicate peaks of BDR1-GFP and tagRFP-ROP11 intensity, respectively. Note that the BDR1-GFP peak overlaps with those of tagRFP-ROP11 and tagRFP-ROP11G17V but not with that of tagRFP-ROP11T22N. **d** Reconstitution of active ROP11 domains in leaf epidermal cells. ROP11, tagRFP-ROPGEF4PRONE, and ROPGAP3 were co-expressed. Either or both BDR1-ECFP and EYFP-WALΔN were co-expressed alongside. Intensity profiles along the white lines are shown at the right panels. Note that EYFP-WALΔN was present around ROPGEF4 clusters in the presence of BDR1-ECFP but not in the absence of BDR1-ECFP. **e** Schematic illustration of ROP signaling pathways in the pits. Activated ROP recruits BDR to the boundary of the pit membrane (red dotted circle), which in turn recruits WAL. WAL promotes assembly of actin microfilaments. The actin bundle directs secondary cell wall ingrowth to overarch the pit membrane (ROP-BDR-WAL-actin pathway). Besides the BDR pathway, activated ROP recruits the MIDD1-Kinesin-13A (KIN13A) complex to the entire area of pit membrane (yellow dotted circle) to promote disassembly of cortical microtubules (MT), thereby inhibiting deposition of secondary cell walls (ROP-MIDD1-Kinesin13A pathway). Bars = 50 μm (**a**) and 10 μm (**c** and **d**). Source data are provided as a Source Data file

**Plant materials and growth conditions**. All *A. thaliana* plants used in this study were in the Col-0 ecotype background. T-DNA insertion lines *wal* (SAIL_729_H08) and *bdr1-1* (SAIL_311_G09) were obtained from ABRC. Seedlings were grown on 1/2 MS agar medium containing 0.2% sucrose or in soil at 22 °C under constant light.

For the expression of estrogen-inducible genes, 5-day-old seedlings were transferred onto 1/2 MS agar medium containing 2 μM estradiol and incubated for 2 days.

**Plasmid construction**. To generate *LexA:GFP-WAL, LexA:tagRFP-WAL, LexA: EYFP-WAL, LexA:WAL-GFP, LexA:BDR1-GFP, LexA:BDR1-ECFP*, and their truncated derivatives, the coding sequences of *WAL* (AT1G58070) and *BDR1* (AT5G06610) were PCR-amplified with the appropriate primers (Supplementary Table 1) and cloned into the pENTR/D-TOPO entry vector (Thermo Scientific). Truncated derivatives were generated by inverted PCR. Clones were recombined with the vectors pER-GX, pER-RX, pER-YX, pER-XG, and pER-XC[18] using LR Clonase Mix II (Thermo Scientific) to form N-terminal GFP, tagRFP, EYFP, C-terminal GFP, and ECFP fusions, respectively.

To generate *pWAL:GFP-WAL* and its truncated derivatives, the 2 kbp region upstream of the translation initiation site of *WAL* and its genomic fragments fused with N-terminal GFP were PCR-amplified, and then fused and cloned into the pEntr/D-TOPO vector using an In-Fusion HD cloning kit (Clontech). These clones were recombined with the pGWB501 vector[32].

To generate *pBDR1:BDR1-GFP, pBDR1:BDR1-tagRFP, pMIDD1:MIDD1ΔN-GFP*, and *pMIDD1:MIDD1ΔN-tagRFP*, the genomic fragments of *BDR1* or *MIDD1* including the 2 kbp region upstream of the translation initiation site were PCR-amplified and cloned into the pEntr/D-TOPO vector. To generate *pMIDD1:Lifeact-GFP* and *pMIDD1:Lifeact-tagRFP*, the 2 kbp region upstream of the translation initiation site of *MIDD1* was PCR-amplified and inserted at the *NotI* site of the pEntr/D-TOPO vector harboring the coding sequence of *Lifeact*. These clones were recombined with the pGWB504 and pGWB559 vectors for C-terminal GFP and tagRFP fusions, respectively[32].

To generate *LexA:BDR3RNAi*, the first 400 bp of the *BDR3* coding sequence and an intron of *WRKY33* were amplified and fused via PCR. The fused sequence was cloned into the pEntr/D-TOPO vector. A 400 bp fragment in the reverse orientation of the *BDR3* coding sequence was inserted into the 3′ end of the *WRKY33* intron using an In-Fusion HD cloning kit. This inverted repeat was recombined with the pMDC7 vector.

For the BiFC assay, entry clones were recombined with pER-YN-X, pER-X-YN, and pER-X-YC[33] to generate an N-terminal fusion with the N-terminal half of EYFP, a C-terminal fusion with the N-terminal half of EYFP, and a C-terminal fusion with the C-terminal half of EYFP, respectively.

**Confocal microscopy**. Cells were observed using an inverted fluorescence microscope (IX83-ZDC, Olympus) fitted with a confocal unit (CSU-W1, Yokogawa), a cooled charge-coupled device (CCD) camera (ORCA-R2, Hamamatsu Photonics), a UPLANSAPO ×60 water-immersion objective (NA = 1.20, Olympus), and laser lines set at 458, 488, and 561 nm. Images were acquired using MetaMorph (Molecular Devices).

To analyze the structure of pits, intact roots were stained with safranin, and thereafter imaged using an Olympus FV3000 inverted confocal microscope equipped with a UPLANSAPO ×100 oil-immersion objective (NA = 1.40, Olympus). FV-OSR software (Olympus) was used to obtain high-resolution images.

**FRET analysis**. Cells were imaged using an Olympus FV3000 inverted confocal microscope equipped with a UPLANSAPO ×60 water-immersion objective (NA = 1.20, Olympus). Bleaching of YFP was achieved with a 514 nm laser at 100% power with 10 iterations. CFP emission was recorded before and after bleaching of YFP. FRET efficiency was calculated as $E = 100(A - B)/B$, where $B$ and $A$ indicated the average intensity of the three images taken before and after bleaching, respectively. The background was subtracted from each image before the calculation[18].

**Electron microscopy**. *Arabidopsis* roots were cut into 3 mm lengths and fixed with 2% (vol/vol) glutaraldehyde in 50 mM sodium cacodylate buffer (pH 7.4) overnight at 4 °C. After washing with the same buffer, preparations were post-fixed with 1% (wt/vol) osmium tetroxide in 50 mM cacodylate buffer for 2 h at 23 °C. Samples were then dehydrated in a graded methanol series (25, 50, 75, 90, and 100% (vol/vol)), infiltrated with increasing concentrations of Epon812 resin (TAAB) (propylene oxide:Epon812 = 3:1, 1:1, 1:3, and 100% (vol/vol)], and embedded. Thin sections (100 nm) were cut with a diamond knife (Diatome) on an ultra-microtome (Leica EM UC7) and mounted on glass slides. Sections were stained with 0.4% (wt/vol) uranyl acetate followed by lead citrate solution and coated with osmium tetroxide for 1 s using an osmium coater (Vacuum Device, HPC-1SW). The slides were observed with an yttrium aluminum garnet backscattered electron (YAG-BSE) detector (10 kV) by a cold field emission scanning electron microscope (Hitachi SU 8220).

**Protein expression and purification**. To generate 6× His-fused BDR1 and GST-fused full-length and truncated WAL, the coding regions of *BDR1* and *WAL* were recombined with pDEST15 and pDEST17 (Thermo Scientific), respectively. The constructs were transformed into *Escherichia coli* BL21-CodonPlus(DE3)-RILP (Agilent Technologies). A single colony was selected and cultured overnight at 37 °C in 3 ml of Luria-Bertani (LB) medium supplemented with 50 mg/l carbenicillin. An aliquot of starter culture (500 μl) was transferred to 50 ml of LB medium and incubated at 37 °C until the $OD_{600}$ reached ~0.4, at which point protein expression was induced with 0.2 mM isopropyl-b-D-thiogalactopyranoside. After 4 h of culture, cells were harvested by centrifugation at $3000×g$ for 2 min.

To purify 6× His-fused proteins, the cell pellet was resuspended in 1.5 ml of His-extraction buffer (50 mM Tris-HCl, pH 7.5, 500 mM NaCl, 20 mM imidazole, 10 mM ATP) and sonicated. The cell lysate was centrifuged at $3000 × g$ for 2 min, and the supernatant was incubated with 150 μl of Ni Sepharose 6 Fast Flow resin (GE Healthcare) at 4 °C for 30 min. The resin was washed and centrifuged three times at $500 × g$ for 1 min with 1 ml of His-extraction buffer. Recombinant proteins were eluted with 500 μl of His-elution buffer (50 mM Tris-HCl, pH 7.5, 500 mM NaCl, 500 mM imidazole). Purification of GST-fused proteins was performed as described above with the exception of using GST-extraction buffer (50 mM Tris-HCl, pH 7.5, 150 mM NaCl), glutathione sepharose 4B resin (GE Healthcare), and GST-elution buffer (50 mM Tris-HCl, pH 7.5, 150 mM NaCl, 20 mM reduced glutathione). Imidazole and reduced glutathione in the eluate were removed using a PD MiniTrap G-25 column (GE Healthcare) with Tris buffer (50 mM Tris-HCl, pH 7.5, 150 mM NaCl). The final products were quantified by SDS-PAGE and Coomassie Brilliant Blue staining using BSA as a standard. Purified proteins were divided into small aliquots, frozen in liquid nitrogen, and stored at −80 °C.

**Pull-down assay**. For pull-down assays, 1 μg of recombinant proteins was mixed and incubated at room temperature for 10 min. Ni Sepharose 6 Fast Flow resin (50 μl) was added to the mixture and incubated for 10 min. To remove unbound proteins, the resin was washed and centrifuged twice with 1 ml of Tris buffer. The remaining resin was then boiled with 50 μl of sample buffer at 95 °C for 3 min. Pull-down products were analyzed by SDS-PAGE and western analysis using mouse monoclonal anti-His-tag antibody OGHis (1:5000; D291-3S; MBL) and rabbit polyclonal anti-GST antibody (1:500; ab9085; Abcam) for detection of ×6 His-fused proteins and GST-fused proteins, respectively.

**Transient expression in *N. benthamiana***. For transient expression in *N. benthamiana*, *R. radiobacter* strains GV3101 MP90 lines harboring different expression constructs were grown in LB media with appropriate antibiotics, harvested by centrifugation at $1800 × g$ for 10 min, and resuspended in infiltration buffer (10 mM MES (pH 5.7), 10 mM $MgCl_2$, 50 mg l$^{-1}$ acetosyringone). Cultures were

adjusted to an OD$_{600}$ of 1.0 and incubated at room temperature. Equal volumes of cultures containing different constructs were mixed for co-infiltration, and then mixed with a culture of *R. radiobacter* carrying the p19 silencing suppressor in a 1:1 ratio. The resulting cultures were infiltrated into leaves of 3- to 4-week-old *N. benthamiana* plants. Leaf samples were harvested 2 days after infiltration and inoculated for 1 day with 2 μM estradiol prior to observation[18].

**Inhibitor treatment**. To transiently depolymerize actin microfilaments, 7-day-old wild-type seedlings were cultured on MS agar medium supplemented with 30 μM latrunculin B (Wako, 10 mM stock in DMSO) for 6 h. To examine the effect of actin disruption on the shape of secondary cell wall pits, 5-day-old wild-type seedlings were cultured on MS agar medium supplemented with 3 μM latrunculin B for 2 days.

**Quantitative reverse transcription PCR analysis**. Total RNA was prepared from *Arabidopsis* cells using the SDS-phenol method, treated with DNase, and purified using an RNeasy Plant Mini kit (QIAGEN). After reverse transcription with oligo (dT)20 priming and SuperScript III reverse transcriptase, quantitative reverse transcription PCR was performed using a LightCycler 96 Instrument (Roche Diagnostics) with FastStart Essential DNA Green Master (Roche Diagnostics). *UBQ10* expression was used as a control. Quantification of gene expression was performed using LightCycler software, and expression levels were determined relative to *UBQ10*.

**Quantification of secondary cell wall pit shape and structure**. To determine the area and aspect ratio of secondary cell wall pits, DIC images of metaxylem vessels were analyzed using ImageJ (https://imagej.nih.gov/ij/). Secondary cell wall pit regions were manually selected, and the area, length, and width were determined using the analyze particle function. The aspect ratio was calculated as the length divided by the width. To determine pit aperture and pit membrane widths, confocal cross-sections of metaxylem vessels were analyzed using ImageJ.

**Quantification of pit-localizing signals**. To evaluate the localization of fluorescent WAL, BDR1, and actin microfilaments in metaxylem vessels, signals on secondary cell wall pits were compared with those under secondary cell walls. Rectangular areas of ~3 × 2 μm were manually selected, and then the average intensity of each area was acquired using ImageJ software.

**Statistical analysis**. Statistical analysis was performed using Student's *t*-test and analysis of variance with Scheffe test. Sample size (*n*) and *p*-value are indicated in the figure legends.

**Expression data analysis**. The expression data of *BDR* genes and *IRX3* were obtained from TRAVA database with default setting (http://travadb.org)[24]. The heatmap was produced with Heatplus R package in Bioconductor (https://www.bioconductor.org/install/).

**Phylogenetic tree construction**. For phylogenetic analysis, amino acid sequences of WAL and BDR homologs were identified using pBLAST against the Phytozome and Spruce Genome Project databases (http://congenie.org/). The phylogenetic tree was constructed using the neighbor-joining method in Molecular Evolutionary Genetics Analysis (MEGA) software. Statistical support for internal branches by bootstrap analyses was calculated using 1000 replications. TAIR accession codes are as follows; WAL-AT1G58070; BDR1-AT5G06610; BDR2-AT1G79420; BDR3-AT1G27690; BDR4-AT3G19540; BDR5-AT1G49840; BDR6-AT5G05840; BDR7-AT3G55720; BDR8-AT5G66740; BDR9-AT1G75160.

**Reporting summary**. Further information on experimental design is available in the Nature Research Reporting Summary linked to this article.

## Data availability

The source data underlying Figs. 1c, e, h, 2f, h, 3d, g, i, and 4b, and Supplementary Figure 3d, f, and i are provided as a Source Data file. The authors declare that all other data supporting the findings of this study are available within the manuscript and its supplementary files or are available from the corresponding author upon request.

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

## Acknowledgements

We thank N. Chua (Rockefeller University) for the pER8 vector, U. Grossniklaus (University of Zurich) for the pMDC7 vector, and T. Nakagawa (Shimane University) for the pGWB vectors. We also thank Y. Noguchi (National Institute of Genetics) and F. Hasegawa (National Institute of Genetics) for technical assistance. This work was supported by Grants in-Aid from the Ministry of Education, Culture, Sports, Science, and Technology of Japan (grants 16H01247 and 15H01243 to Y.O. and 15H05958 to H.F.), the Japan Society for the Promotion of Science (grants 16H06172 and 18H02469 to Y.O. and 16H06377 to H.F.), the Japan Science and Technology Agency (Precursory Research for Embryonic Science and Technology project to Y.O.; Grant JPMJPR11B3), the National Institute of Genetics (NIG-JOINT2015-A1-26 to H.F.), and the Mitsubishi Foundation to Y.O.. Y.S. and Y.N. are special joint researchers of the National Institute of Genetics.

## Author contributions

Y.O. and H.F. designed the research. Y.S. and Y.N. performed the experiments. M.W., M.S., and K.T. performed electron microscopy. Y.S., Y.N., H.F., and Y.O. wrote the manuscript.

## Additional information

**Competing interests:** The authors declare no competing interests.

