## [Peer Review File · Nature Communications]

Reviewers' comments:

Reviewer #1 (Remarks to the Author):

In this paper Sugiyama et al has identified a new ROP signaling pathway that is involved in the establishment of the pit-cell wall boundary in Arabidopsis xylem cells. They show that this pathway (BDR1/3-WAL) promotes the formation of F-actin ring in this boundary and is important for controlling the pit pattern in the xylem secondary cell wall. The findings are quite interesting and significant, and they provide insights into the unknown mechanism for the formation of the pit-cell wall boundary in these cells, complement the previous on the ROP11-MIDD1 pathway and expand the list of ROP GTPase signaling pathways. Overall the data are quite solid and well analyzed, and the manuscript is clearly written in a concise manner for general audience. I support the publication of this work in NC after the following minor issues are addressed.

We can be confident of the conclusion that both the MIDD1 and BDR1/3 pathways are regulated by ROPs, but it is inconclusive whether they are activated by the same ROP, ROP11. The interaction and localization assays were performed using dominant mutant forms of ROP11 or in a heterologous system. To unequivocally demonstrate that it is ROP11, but not another ROP, that activates the BDR1-WAL pathway, one would need no loss of ROP11 function mutants to analyze BDR1/WAL localization and in vivo co-IP to confirm the formation of ROP11, BDR1/3 and WAL complex. For example, ROP2/4 and ROP6 are closely related, but they activate RIC4 and RIC1 to promote the organization of cortical actin and microtubule, respectively, although both constitutively active ROP2 and ROP6 interact with RIC1 and RIC4. Unless the authors can include relevant data, it would suggest that the authors be cautious in making the conclusion.

My second point, which might be connected to the above issue, is that it may be worthy of discussing the two opposing ROP signaling pathways for the regulation of secondary cell wall patterns (the MIDD1 pathway for pit formation and the BDR1 pathway for restricting the pit) in the broader content of ROP signaling that regulate cell morphogenesis and cell wall patterning. There seems to be an emerging theme in this field. Leaf pavement cell morphogenesis is regulated by two opposing but complementary ROP signaling pathways: the ROP2/4-RIC4-actin and the ROP6-RIC1-microtubule pathway (Fu et al. 2005. Cell; Fu et al. 2009. Cur Bio). Similarly a paper that will be published in Nat Plant identified a new ROP-microtubule pathway localized to the shank of root hairs, again opposing and complementary to the ROP2-actin pathway localized to the tip of root hairs. Accordingly, the cited literature should be expanded. Both if the cell wall field and ROP signaling field are very active, but only a total of 18 references was cited, of which a large portion was from the work by the senior author.

The title is bit too general and lacks accuracy for the reasons discussed above.

In the main text, there was no description of how the BDR family was identified.

Reviewer #2 (Remarks to the Author):

The study characterizes the novel ROP11-BDR1-WAL-actin signaling pathway that controls bordered cell wall formation at boundaries of pits, involving two new players in pit formation during secondary vascular development. The respective proteins WAL and BD are upregulated in this process and show pit-specific localization
WAL is an uncharacterized protein without known domains. It binds to actin directly and stabilizes actin filaments. In ^{wal} mutants a characteristic actin ring at the pit border is missing leading to expanded pit areas. Treatment with LatB causes similar phenotypes further

corroborating the role of WAL for actin stabilization at the pit border.

BDR1 is a direct interactor of WAL, co-localizes with WAL and is similarly involved in the pit-shaping process. BDR1 together with BDR3 are required for recruitment of WAL, but not vice versa. BDR1 is an effector of ROP11.

The findings are original, identifying novel components of ROP signaling and shedding light on the role of actin in pit formation, which has not been investigated so far. These results will be of interest to a broad audience as Rho/Rac small GTPase signaling is involved in a vast number of cellular processes that require regulation and organization of the cytoskeleton. The identification of these new effectors will certainly spark the characterization of orthologs in different signaling pathways. The work appears technically sound and statistics are provided where appropriate. The manuscript is well written and results and reasoning easy to follow and well supported and I enjoyed reading it.

Reviewer #3 (Remarks to the Author):

This paper is a very nice outline of how actin rings and associated proteins aid in the formation of pits are formed during meta-xylem development. The authors identify two protein families that contribute to this function, referred to as WAL and BDR, and outline how these are recruited by active ROP11 to the pit membranes and possibly to define the boundary of the pits.

I think this is a very nicely executed study and I only have two major remarks for the authors:

1. It would be nice to show that the BDR1-GFP construct is functional (I failed to see this in the paper). I assume that this can be done by introducing the construct into the *bdr1 BDR3-RNAi* line?
2. It was not clear to me why the truncated WAL protein construct was used in the ROP11 recruitment assay rather than the full-length version. Could the authors elaborate on this??

Apart from this, I think the data looks very nice.

Minor things:

The abstract reads a bit weird and might be improved through the read/revisions of a native english speaker.

IRX3 might need an explanation.

"Novel" in abstract is not needed...the paper is good enough without this claim.

Response to Reviewers' comments

Reviewer #1:

In this paper Sugiyama et al has identified a new ROP signaling pathway that is involved in the establishment of the pit-cell wall boundary in Arabidopsis xylem cells. They show that this pathway (BDR1/3-WAL) promotes the formation of F-actin ring in this boundary and is important for controlling the pit pattern in the xylem secondary cell wall. The findings are quite interesting and significant, and they provide insights into the unknown mechanism for the formation of the pit-cell wall boundary in these cells, complement the previous on the ROP11-MIDD1 pathway and expand the list of ROP GTPase signaling pathways. Overall the data are quite solid and well analyzed, and the manuscript is clearly written in a concise manner for general audience. I support the publication of this work in NC after the following minor issues are addressed.

Answer: Thank you for your positive evaluation of our manuscript.

We can be confident of the conclusion that both the MIDD1 and BDR1/3 pathways are regulated by ROPs, but it is inconclusive whether they are activated by the same ROP, ROP11. The interaction and localization assays were performed using dominant mutant forms of ROP11 or in a heterologous system. To unequivocally demonstrate that it is ROP11, but not another ROP, that activates the BDR1-WAL pathway, one would need no loss of ROP11 function mutants to analyze BDR1/WAL localization and in vivo co-IP to confirm the formation of ROP11, BDR1/3 and WAL complex. For example, ROP2/4 and ROP6 are closely related, but they activate RIC4 and RIC1 to promote the organization of cortical actin and microtubule, respectively, although both constitutively active ROP2 and ROP6 interact with RIC1 and RIC4. Unless the authors can include relevant data, it would suggest that the authors be cautious in making the conclusion.

Answer: We agree that additional experiments are required to demonstrate that BDR1/3 are regulated exclusively by ROP11. IP of native ROPs and genetic analysis of ROP family proteins would be a reasonable approach to clarify this issue. However, these experiments are technically challenging and would take years to complete, because the differentiating xylem vessel cell population in plant tissue is very small and the ROP family is highly redundant. Thus, we think that determining the complete set of ROPs regulating BDR1/3 is beyond the scope of this study. However, we have modified the Title, Abstract, and

Discussion to present a more cautious conclusion. In the revised manuscript, we no longer claim that ROP11 regulates both BDR1 and MIDD1, or that the ROP signal diverges into two signals. We have explained that ROP11 is one of many possible ROPs regulating BDR1/3 and that other ROPs may regulate BDR1/3 redundantly or independently of ROP11. We conclude that two ROP signaling pathways that oppositely regulate cell wall growth would allow pit formation.

My second point, which might be connected to the above issue, is that it may be worthy of discussing the two opposing ROP signaling pathways for the regulation of secondary cell wall patterns (the MIDD1 pathway for pit formation and the BDR1 pathway for restricting the pit) in the broader content of ROP signaling that regulate cell morphogenesis and cell wall patterning. There seems to be an emerging theme in this field. Leaf pavement cell morphogenesis is regulated by two opposing but complementary ROP signaling pathways: the ROP2/4-RIC4-actin and the ROP6-RIC1-microtubule pathway (Fu et al. 2005. Cell; Fu et al. 2009. Cur Bio). Similarly a paper that will be published in Nat Plant identified a new ROP-microtubule pathway localized to the shank of root hairs, again opposing and complementary to the ROP2-actin pathway localized to the tip of root hairs. Accordingly, the cited literature should be expanded. Both if the cell wall field and ROP signaling field are very active, but only a total of 18 references was cited, of which a large portion was from the work by the senior author.

Answer: We have expanded the Discussion by adding additional references including references to the three papers you suggested. We have also added related references to the Introduction. In all, 34 references are cited in the manuscript, including in Supplementary Information.

The title is bit too general and lacks accuracy for the reasons discussed above.

Answer: We modified the Title. The new title of our revised manuscript is “A Rho-actin signaling pathway shapes cell wall boundaries in Arabidopsis xylem vessels”.

In the main text, there was no description of how the BDR family was identified.

Answer: We have added an explanation about how we identified the BDR family. As we did for WAL, we further screened for xylem-expressed gene products in cultured xylem cells and found a protein, BDR1, that localized to pit boundaries.

Reviewer #2:

The study characterizes the novel ROP11-BDR1-WAL-actin signaling pathway that controls bordered cell wall formation at boundaries of pits, involving two new players in pit formation during secondary vascular development. The respective proteins WAL and BD are upregulated in this process and show pit-specific localization. WAL is an uncharacterized protein without known domains. It binds to actin directly and stabilizes actin filaments. In wal mutants a characteristic actin ring at the pit border is missing leading to expanded pit areas. Treatment with LatB causes similar phenotypes further corroborating the role of WAL for actin stabilization at the pit border.

BDR1 is a direct interactor of WAL, co-localizes with WAL and is similarly involved in the pit-shaping process. BDR1 together with BDR3 are required for recruitment of WAL, but not vice versa. BDR1 is an effector of ROP11.

The findings are original, identifying novel components of ROP signaling and shedding light on the role of actin in pit formation, which has not been investigated so far. These results will be of interest to a broad audience as Rho/Rac small GTPase signaling is involved in a vast number of cellular processes that require regulation and organization of the cytoskeleton. The identification of these new effectors will certainly spark the characterization of orthologs in different signaling pathways. The work appears technically sound and statistics are provided where appropriate. The manuscript is well written and results and reasoning easy to follow and well supported and I enjoyed reading it.

Answer: Thank you for your positive evaluation of our manuscript.

Reviewer #3:

This paper is a very nice outline of how actin rings and associated proteins aid in the formation of pits are formed during meta-xylem development. The authors identify two protein families that contribute to this function, referred to as WAL and BDR, and outline how these are recruited by active ROP11 to the pit membranes and possibly to define the boundary of the pits.

I think this is a very nicely executed study and I only have two major remarks for the authors:

Answer: Thank you for your positive evaluation of our manuscript.

1. It would be nice to show that the BDR1-GFP construct is functional (I failed to see this in the paper). I assume that this can be done by introducing the construct into the *bdr1 BDR3-RNAi* line?

Answer: Thank you for this suggestion. It is really an excellent idea. To determine whether the BDR1-GFP construct is functional or not, we introduced *pBDR1:BDR1-GFP* into the *bdr1-1 BDR3RNAi* line. The results showed that the *BDR1-GFP* rescued the pit phenotype of the mutant, confirming that BDR1-GFP was functional. We have added the data to the manuscript (Supplementary Figure 3e and f).

2. It was not clear to me why the truncated WAL protein construct was used in the ROP11 recruitment assay rather than the full-length version. Could the authors elaborate on this??

Answer: In the recruitment assay (Fig. 4d), our aim was to determine whether BDR recruits WAL to the plasma membrane, independently of the WAL-actin interaction. WAL Δ N possesses the BDR1-interacting region but lacks the actin-binding region of WAL. Thus, WAL Δ N was the best construct to test for WAL recruitment by BDR. We have added an explanation about this to the text.

Apart from this, I think the data looks very nice.

Answer: Thank you for your positive evaluation of our data.

Minor things:

The abstract reads a bit weird and might be improved through the read/revisions of a native English speaker.

Answer: The manuscript has been edited again by professional native speakers.

IRX3 might need an explanation.

Answer: We have added an explanation of *IRX3* to the legend of Supplementary Figure 3c. *IRX3* encodes a cellulose synthase complex subunit that is expressed specifically in xylem cells. Thus, *IRX3* was used as a marker for xylem development.

"Novel" in abstract is not needed...the paper is good enough without this claim.

Answer: Thank you for your suggestion. We have removed "novel" from the Abstract.